# Association of plasma miRNAs with early life performance and aging in dairy cattle

**Madison MacLeay[1], Georgios Banos[1,2], Francesc Xavier Donadeu[1] ***

**1** The Roslin Institute and Royal (Dick) School of Veterinary Studies, University of Edinburgh, Midlothian, United Kingdom, **2** Department of Animal and Veterinary Sciences, Scotland's Rural College (SRUC), Roslin Institute Building, Midlothian, United Kingdom

* xavier.donadeu@roslin.ed.ac.uk

**Data Availability Statement:** All relevant data are within the manuscript and its Supporting information files.

**Funding:** MM received a University of Edinburgh Principal's Career Development PhD Scholarship

## Abstract

Early life performance traits in dairy cattle can have important influences on lifetime productivity. Poor health and fertility are of great economical and animal welfare concern. Circulating miRNAs have been linked to several livestock traits, including resistance to infection, fertility, and muscle development. This study aimed to identify circulating miRNAs associated with early life performance traits and aging in dairy cattle. Plasma samples from female calves (n = 12) identified retrospectively as differing in health, growth, and fertility outcomes prior to first calving were analyzed using PCR arrays detecting 378 miRNAs. Levels of 6 miRNAs differed significantly in calves with poor growth/fertility relative to controls (t-test: P<0.05). Additionally, general(ized) (non)linear mixed models identified 1 miRNA associated with average daily gain until weaning, 22 with live bodyweight at one year of age, 47 with age at first service, and 19 with number of infections before first calving. Out of 85 distinct miRNAs that were associated with at least one animal trait, 9 miRNAs were validated by RT-qPCR in a larger cohort (n = 91 animals), which included longitudinal plasma samples (calf, heifer, first lactation cow). Significant associations (P<0.05) involving individual miRNAs or ratios between miRNAs and early-life performance traits were identified, but did not retain significance after multiple testing adjustment. However, levels of 8 plasma miRNAs (miR-126-3p, miR-127, miR-142-5p, miR-154b, miR-27b, miR-30c-5p, miR-34a, miR-363) changed significantly with age, most prominently during the calf-to-heifer transition. Comparative RT-qPCR analyses of these miRNAs across 19 calf tissues showed that most were ubiquitously expressed. Online database mining identified several pathways involved in metabolism and cell signaling as putative biological targets of these miRNAs. These results suggest that miR-126-3p, miR-127, miR-142-5p, miR-154b, miR-27b, miR-30c-5p, miR-34a, miR-363 are involved in regulating growth and development from birth to first lactation (~2 years old) and could provide useful biomarkers of aging in cattle.

## Background

Disease and infertility in modern dairy herds may lead to a significant reduction in milk production and an increase in involuntary culling, thus reducing profit margin. About 20% of cows in a herd are culled prior to the first lactation [1], with highest losses seen in the first day

(https://www.ed.ac.uk/institute-academic-development/postgraduate/doctoral/career-management/principals-scholarships#:~:text=Holders%20of%20a%20Principal's%20Career,e.g.%20in%20Schools%20or%20Centres)). The funders had no role in study design, data collection and analysis, decision to publish, or preparation of the manuscript.

**Competing interests:** The authors have declared that no competing interests exist.

of life (mean 7.9% of animals in a herd) and in heifers that do not conceive or abort (mean 4.2%) [2, 3]. Animals usually become profitable only during the second lactation; thus, most losses occur before profits are seen, with disease and infertility being major causes [1]. Because of this, there has been an increasing interest in selection for enhanced functional traits such as disease resistance, fertility, and growth, which begin to manifest in early life. This interest is compounded by rising policymaker and consumer demand for healthier animals with better welfare and less exposure to medication (e.g., antibiotics) [4, 5].

Improvements in early performance of functional traits are likely to affect long-term health and longevity, as well as milk production and therefore, lifetime productivity. For example, rapid growth can lead to earlier puberty and younger age at first service, hence, earlier onset of milk production. Growth rate has been associated with fertility, longevity, milk yield, and protein and fat content of milk [6–8]. For these reasons, growth rate has been considered as a predictor of fertility, and can be improved by nutrition management [6]. Growth and body condition also impact on pregnancy rates and time to return to estrus after calving [9, 10]. Good heifer fertility, manifested by early estrus and high conception rate, is associated with earlier lactation, and improved conception rates and milk yields in subsequent pregnancies [2]. Young animals also exhibit the highest culling rates. Disease is the leading cause of mortality for calves under 6 months [11]; common diseases in this age group include scours (diarrhea) and pneumonia.

Functional traits have been included in the breeding goals of selection programs in recent years, however, progress has been slow due to low heritability of health and fertility, and the large number of traits being selected at once [4]. Negative genetic correlations also contribute to worsening fertility [12] and disease [13] when increased milk production is being selected for [14, 15].

miRNAs are small (~22nt), noncoding RNA molecules that regulate post-transcriptional gene expression, playing important roles in fine-tuning many cellular processes in humans and animals. miRNAs secreted by body tissues are relatively stable due to their association with vesicles, proteins, or lipids [16], and are relatively easy to quantify in blood and other biofluids. In humans and model species, miRNAs are natural key mediators of cellular responses to pathophysiological stressors and aging [17]. Because of this, circulating miRNAs can provide useful biomarkers for assessing tissue fitness and resilience, as well as disease risk. In humans, changes in miRNA profiles in circulation in middle aged and older adults were associated with individual risk for cardiovascular disease [18], metabolic syndrome/type 2 diabetes [19, 20], cancer, and lifespan [21]. These findings are consistent with data in phylogenetically distant species such as nematodes [22].

miRNAs have been linked to health and production traits in livestock. For example, miR-1 and miR-206 were associated with the double-muscle phenotype in Texel sheep [23], and other miRNAs were suggested as biomarkers of puberty in chicken [24], and of estrus cycle stage and pregnancy in cattle [25, 26]. Associations between miRNA levels and disease have also been reported in livestock (see [27] for a review). For example, distinct miRNA profiles have been reported in cattle after exposure to the causative agent of Johne's disease [28] and during clinical metritis [29]. Several studies have reported miRNA profiles associated with mastitis, although they can vary with the pathogen involved and the type of tissue or biofluid analyzed [30–32]. However, the associations between circulating miRNA levels and overall disease risk in livestock have not been investigated.

Age can also have a major impact on miRNA expression profiles. We previously identified age-associated miRNAs in Holstein-Friesian cattle, and reported miRNAs associated with functional traits in adult cows, such as fertility, lameness, and milk yield and content [33]. Similarly, in sheep various miRNAs were related to age and other traits such as body and carcass

mass [34]. Moreover, Tewari et al. [35] identified several differentially expressed plasma miR-NAs in four different age groups of Piedmontese cattle, from newborn up to 15–17 months, and Li et al. identified 126 differentially expressed miRNAs in adipose tissue between neonatal and adult pigs [36]. However, none of those studies involved longitudinal measurements within the same animal, thus preventing inferences on the association between miRNA profiles and animal performance.

The objective of this study was to assess the association between circulating miRNA profiles, and early life performance traits and aging in dairy cattle using, for the first time, longitudinal measurements on animals. Finding such associations could provide novel early-life predictors of health and productivity for the benefit of the dairy industry.

## Methods

### Experimental animals and sample collection

All animal procedures were performed with approval from The Roslin Institute (University of Edinburgh) Animal Welfare and Ethical Review Board and following the UK Animals (Scientific Procedures) Act, 1986. This study was conducted and reported in accordance with ARRIVE guidelines.

Blood samples were collected from 227 female calves aged ≤1 month of Holstein-Friesian, Brown Swiss or Norwegian Red breed or crosses, born at Langhill Farm (University of Edinburgh, Scotland). Samples were collected during two successive calving seasons (September to April) in 2017/18 (n = 119) and 2018/19 (n = 108). Blood samples were again collected on two later dates from the same animals (except those that had been culled or died), as heifers (aged 14–23 months; n = 209) and first lactation cows (aged 29–35 months; n = 171), respectively. In all cases, blood samples were transported on ice for same-day processing, as described [33]. In brief, whole blood was centrifuged at 1,900xg for 10 minutes at 4˚C. The supernatant was pipetted into microcentrifuge tubes and centrifuged at maximum speed for 10 minutes at 4˚C. The supernatant was transferred into fresh tubes and stored at -80˚C for later use. Only samples devoid of hemolysis, based on absorbance at 414nm as measured on a Nanodrop 1000 spectrophotometer [37], were used for subsequent analyses.

Samples from 19 different body tissues were collected from an additional three healthy 5-month-old Holstein-Friesian male calves sacrificed as part of another study at the University of Edinburgh. Tissues were collected into Eppendorf tubes on dry ice and stored at -80˚C until use.

### Identification of candidate miRNAs using PCR array profiling

**Animal selection.** Calf samples (≤1 month of age) to be used for PCR array profiling were selected using early life performance data from all animals (including birth weight, average daily gain until weaning (ADG-wean), live bodyweight at one year of age, average daily gain until first service (ADG-AI), and number of services to first conception (S/C)), to retrospectively identify three groups (n = 4 animals each) with distinctly poor health (Group H), poor growth/fertility (Group G), or controls with good growth/fertility and no reported health issues (Group C). Animals with ill-health, i.e. that contracted infection (primarily pneumonia and scours) and died within the first year of life, were assigned to Group H.

To assign calves to Groups G and C, principal component analysis (PCA) and hierarchical clustering with complete linkage in R version 3.6.2 were performed using the abovementioned early-life animal trait data. Prior to inclusion in the PCA, all animal traits were adjusted for breed, dam age and lactation number, and month and cohort (calving season) of birth; S/C was additionally adjusted for month born and year-one weight. Adjustments were done using

a regression of factors against each trait under consideration, then subtracting each data point's residual from the overall trait mean. After clustering, the animal with the worst overall performance on adjusted performance traits (e.g. lowest growth, latest and most services) and the animal with the best performance considering these traits were identified. The dendrogram produced by the hierarchical clustering function was used to identify 3 animals that had similar performance and were clustered with the worst or best. The four worst-performing individuals were included in Group G, whereas the four best-performing, which also had no known past illnesses, were used as controls (Group C).

**miRNA analyses.** miRNAs were profiled in plasma samples from the 12 animals in Groups H, G, and C using Qiagen services and custom miRCURY LNA PCR arrays in 384-well plates. These included all miRNAs (total, 378) detectable in bovine blood, as established previously [25, 26]. Expression levels were derived from crossing threshold (Ct) levels then normalized using global mean normalization [38]. Data for a particular miRNA were excluded if Ct values were >35 in at least 2 animals in a single group (183 miRNAs excluded, 195 miRNAs analyzed).

PCR array data were separately analyzed using t-tests and general(ized) (non)linear mixed models (GLMMs). T-tests were performed on log-transformed normalized expression data to determine whether miRNA levels differed between Group C and each of Groups G and H. Additionally, GLMMs in ASReml 4 were used to determine whether levels of each miRNA were associated with the traits of interest, i.e., number of infections before calving, age at first service (AFS), S/C, year one weight, and ADG-wean. A linear model was used for normally distributed traits and a nonlinear (Poisson) model was used for number of infections and S/C. The models included fixed effects known to influence each trait, e.g., breed, cohort, etc., which were first determined using a backwards selection process from the maximal model by comparing AIC or, for Poisson distributed models, log likelihood. The final list of terms included in the models are described in S2 Table 1 in S1 File. Animal ID (associated with pedigree to three generations) was always included as a random effect to account for additive genetic effect. Continuous effects were scaled and centered. Finally, the normalized level for each miRNA and the age at sampling (days) were incorporated.

## RT-qPCR validation of miRNA-trait associations

Nine miRNAs were selected for further validation. An extended set of animals was used (n = 91, based on a power calculation using Cohen's f2 [39]). After accounting for animal culling and sample hemolysis a total of 88 calf samples, 57 heifer samples, and 56 cow samples were available for analyses. RT-qPCR was used to validate associations between selected miRNAs and the traits listed above by quantifying the levels of individual miRNAs in plasma samples.

RNA was extracted using TriZol as previously described [40], except that samples were incubated in isopropanol at -20˚C for 1 hr. and eluted into 20μl RNAse-free water. Reverse transcription was performed using the miRCURY LNA RT kit and qPCR using the miRCURY LNA SYBR Green PCR kit and miRCURY LNA miRNA PCR assays (Qiagen), according to the manufacturer's instructions. A Stratagene Mx3000P qPCR system was used (Agilent, USA). Relative transcript levels were calculated from a standard curve made from serial dilutions of a test sample cDNA pool, using the machine software automatic settings. Samples with >1.5 Ct difference between technical replicates were discarded. The mean of two reference miRNAs, bta-miR-20a and bta-miR-106a, was used for normalization of miRNA expression levels. Those two miRNAs were chosen among six candidate miRNAs (bta-miR-20a, bta-miR-106a, bta-miR-19a, bta-miR148a, bta-miR-101 and the spike-in, cel-miR-39-3p) using the geNorm algorithm [41].

QPCR data were analyzed for associations between plasma miRNA levels and early performance traits using the same GLMMs as for the array data above, with the addition of the RT-qPCR batch number as a fixed term to account for technical variation. Expression data for single miRNAs, ratios between pairs of miRNAs in calves, and fold change in miRNA levels from calf to heifer, were analyzed. For ratios, all possible combinations of miRNAs were considered, irrespective of order (36 ratios in total). For fold changes, the heifer miRNA level was divided by the calf miRNA level in the same animal, if both samples were available.

For each analysis, False Discovery Rate (FDR) was used to correct P-values for multiple testing using R (p.adjust function, considering 9 tests involving miRNA levels or miRNA fold changes, and 36 tests involving ratios between miRNAs). We also calculated the power each mixed model had to detect a significant association between miRNA and the trait of interest using Cohen's $f^2$ [39].

### Age-related changes in miRNAs

The associations between the 9 miRNAs above and age were also examined using RT-qPCR data. A general linear mixed model was run for log-transformed normalized miRNA data using R v4.0.4. The independent variables were batch number, sampling age (days, as a continuous variable), and animal ID (random). The effect of age group (as a factor) was also investigated using batch number, age group, and animal ID (random) as independent variables, followed by pairwise comparisons of estimated marginal means. The FDR method was used to adjust p-values (9 tests).

### Tissue expression analyses

Levels of selected miRNAs were quantified in 19 body tissues from each of 3 calves: gallbladder, ileum, semitendinosus muscle, skin, tongue, trachea, esophagus, lung, heart, vena cava, carotid artery, rumen, liver, lymph node, spleen, colon, kidney, testis, and rib cartilage. For quantification of miRNAs in solid tissues, 1ml TriZol was first added to 50mg of each sample and homogenized in Lysing matrix D tubes using a FastPrep homogenizer (MP Biomedicals) at speed 4 for up to 1 minute. The homogenate was removed and centrifuged at 12000xg for 10 minutes at 4˚C and the supernatant transferred to a fresh tube for RNA extraction following the TriZol-based protocol described above. RNA was quantified on a Nanodrop 1000 and diluted to 10ng/μl, followed by RT-qPCR using the same kits and manufacturer's protocol described for plasma. Relative expression levels were determined using a standard curve and normalized to snU6b [42], and the log-transformed normalized miRNA levels were analyzed using a GLMM with tissue (fixed) and animal ID (random) as independent variables in R package nlme (v3.1–152 [43]) with Wald tests (anova.lme). P-values were adjusted using FDR (in R).

### Pathway analysis

miRPath v3.0 was used to identify gene pathways targeted by human homologues of bovine miRNAs of interest. This involved KEGG analysis of data from Tarbase v7.0, with the merging algorithm set to pathways union and other parameters automatically set.

## Results

### Identification of candidate miRNAs using PCR array profiling

For initial identification of miRNA candidates, array profiling of plasma miRNAs was performed in a limited set of animals. As described in Methods, PCA and hierarchical clustering

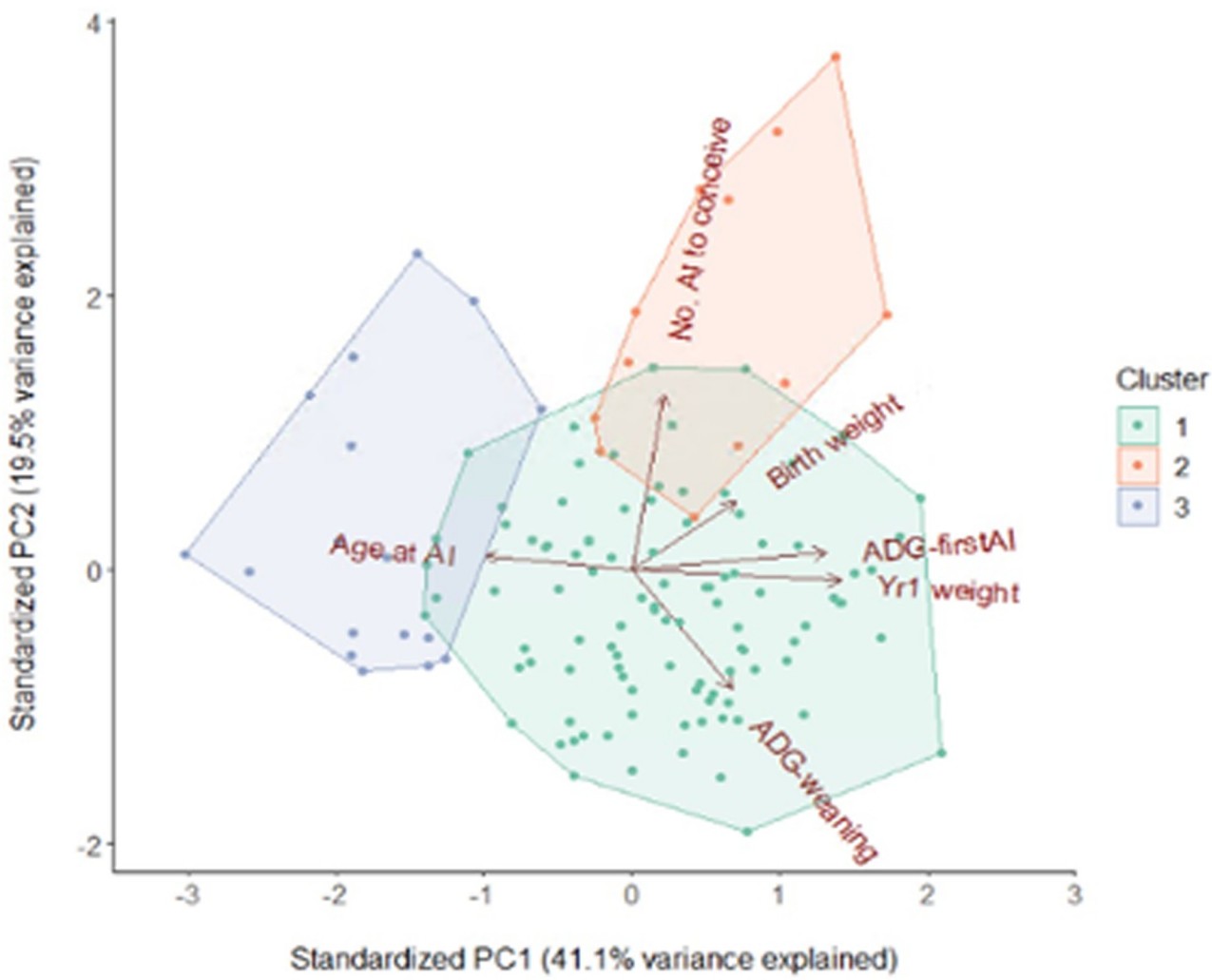

**Fig 1. PCA plot generated using growth/fertility trait data.** Red axes show the direction of the trait (where AI denotes artificial insemination, Age at AI is age at first service, ADG-firstAI is average daily gain until first service, and ADG-weaning is average daily gain until weaning). Hierarchical clustering was used to identify three clusters, indicated by color; this produced a dendrogram used for selecting groups of calves for analysis. Group G included the worst performers from cluster 3, and Group C the best performers from cluster 1.

were used to segregate animals according to their performance on growth and fertility traits. The biplot for the first two principal components is shown in Fig 1, with clusters indicated. These were used to identify animals with poor growth/fertility (Group G) and good growth/fertility (control; Group C). An additional group of animals with distinctly poor health were assigned to Group H, thus the groups represented the most extreme animals in the herd. The performance of each group on the relevant traits is shown in Table 1, demonstrating that animals performed as expected (i.e. higher infection rate in Group H, and lower ADG-wean and one year bodyweight combined with later first service and higher number of services in Group G, compared to Group C).

PCR arrays were used to profile 378 miRNAs in plasma samples from Groups H, G and C. Ct values are provided in Additional Data. Relative miRNA abundance, as determined by the distribution of Ct values, was similar across samples, as shown in Fig 2a. Overall, 92% of miRNAs (353) were detected in at least one sample, 53% were detected in every sample (203

**Table 1. Performance of each group (n = 4 animals) used for PCR array profiling, on the traits of interest (mean (SE)).**

| Trait | Group H | Group G | Group C |
|---|---|---|---|
| ADG-wean (average daily gain until weaning, kg/day) | 0.67 (0.11) | 0.50 (0.06) | 0.85 (0.04) |
| Year 1 weight (kg) | NA | 331.75 (15.58) | 385 (27.23) |
| AFS (age at first service, days) | NA | 505.25 (18.26) | 429 (4.76) |
| S/C (services in first conception) | NA | 2.25 (0.48) | 1.25 (0.25) |
| No. infections | 1.5 (0.29) | 0 | 0 |

miRNAs), and 8% were not detected in any sample (no Ct, 31 miRNAs). A list of the 20 most abundant miRNAs (on average) in all samples is shown in Fig 2b and is consistent with previous data from calf plasma [33].

To investigate whether miRNA levels are associated with the five traits of interest, we used GLMMs and t-tests separately on PCR array data. GLMMs are described in S2 Table 1 in S1 File. In total, 85 distinct miRNAs were found to be significantly associated with a trait of interest (GLMMs, P<0.05) or to be significantly different between groups (t-tests, P<0.05; S2 Tables 2 and 3 in S1 File). From the GLMMs, one miRNA was associated with average daily gain until weaning (ADG-wean), 22 with year 1 weight, 47 with age at first service (AFS), and 19 with number of infections. From the t-tests, 6 miRNAs differed significantly between Groups G and C. However, none of these associations were significant after FDR adjustment (P<0.05).

## Validation of miRNA-trait associations

From the qPCR array dataset 9 different miRNAs were chosen for further validation. miRNAs were chosen if they were both significant for Group in a t-test and significantly associated with a trait in a GLMM; if they were significant in GLMMs and had one of the largest predicted effects; or if they were significant in GLMMs and also known to be associated with age [33]. These miRNAs (miR-126-3p, miR-127, miR-142-5p, miR-154b, miR-27b, miR-30c, miR-34a, miR-363, and miR-425-3p; in bold in S2 Tables 2 and 3 in S1 File) were individually quantified by RT-qPCR using plasma samples from an extended set of animals (n = 91) across three life stages (calf, heifer and 1st lactation cow).

miRNA levels in calves, pairwise miRNA ratios in calves, and the fold-change in miRNA levels between calf and heifer were obtained and analyzed using the same models as before (S2 Table 1 in S1 File). Significant associations were identified between some individual miRNAs and miRNA ratios, and animal traits (Table 2), however, none remained statistically significant after FDR adjustment. miRNA fold changes had no statistical associations with the studied animal traits.

Before p-value adjustment, the models with miRNAs or ratios that were significantly associated with the trait of interest had 19–42% power to detect the significant association (P<0.05).

## Age-related changes in miRNAs

We also investigated how the same nine miRNAs above changed across calf, heifer, and mature cow stages (Fig 3). We found that three miRNAs decreased with age (miR-127, miR-154b, miR-363), five increased with age (miR-142-5p, miR-126-3p, miR-30c-5p, miR-34a, miR-27b), and one (miR-425-3p) did not change across the three stages. We also analyzed the effect of age as a factor and performed pairwise comparisons of the estimated marginal means (Fig 4). As seen in Figs 3 and 4, the largest differences were seen between the calf and heifer stages.

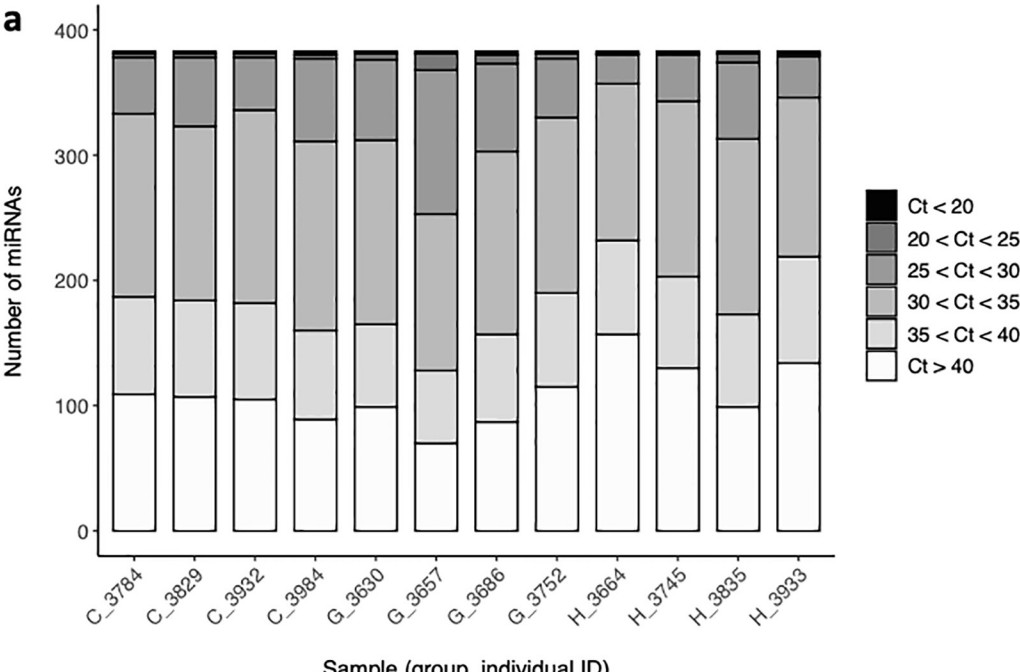

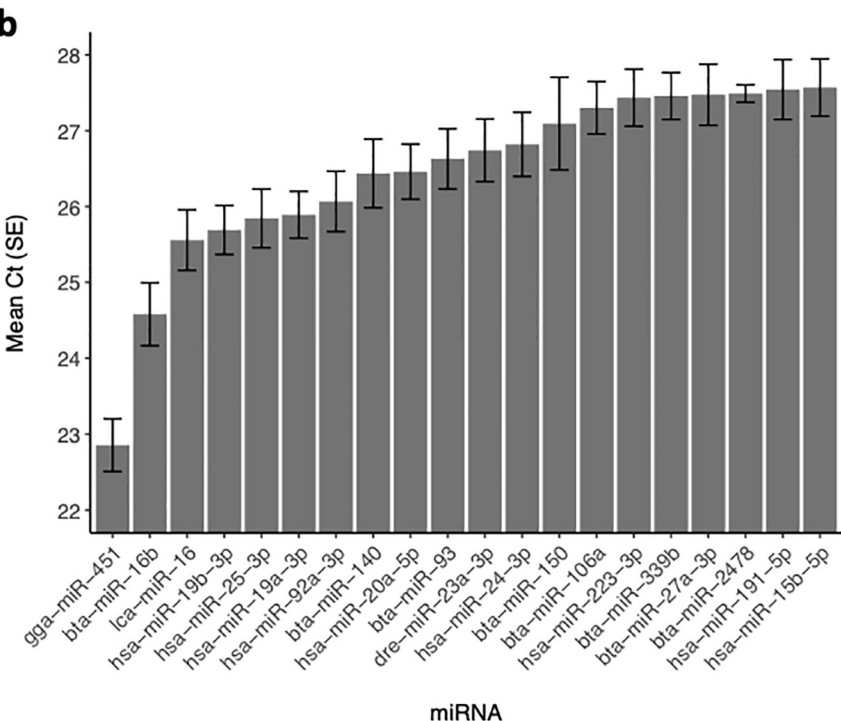

**Fig 2. miRNA detection in calf plasma by PCR arrays: a) number of miRNAs detected at different Cq ranges in each sample and b) top 20 most abundant miRNAs (mean Cq +/- SE), after exclusion of spiked-in sequences.**

**Table 2. Individual miRNA and miRNA ratios that were significantly associated with early life traits prior to FDR adjustment (GLMM: P<0.05).**

| Trait | miRNA or ratio | Predicted Effect (SE or 95% CI)[1] | P-value |
|---|---|---|---|
| ADG-wean (average daily gain until weaning) | miR-30c-5p | 0.03 (0.01) | 0.03 |
| No. infections | miR-126-3p | 1.4 (1.07–1.830) | 0.02 |
| ADG-wean | miR-425-3p: miR-363 | -0.04 (0.01) | 0.01 |
|  | miR-142-5p: miR-363 | -0.04 (0.01) | 0.01 |
| Age 1st service | miR-154b: miR-30c-5p | 10.47 (3.77) | 0.01 |
|  | miR-27b: miR-30c-5p | 7.31 (3.51) | 0.04 |
| No. infections | miR-126-3p: miR-363 | 1.34 (1.05–1.70) | 0.02 |
|  | miR-34a: miR-363 | 1.61 (1.17–2.22) | 0.01 |
|  | miR-363: miR-127 | 0.64 (0.42–0.98) | 0.045 |
|  | miR-363: miR-30c-5p | 0.58 (0.36–0.92) | 0.03 |

[1]For normally distributed traits, predicted effect is additive and SE is given in parentheses. For traits following a Poisson distribution (no. infections), effect is multiplicative, and 95% CI is given.

Where age as a continuous trait was significantly associated with miRNA level, the models had 71–100% to detect the significant association (P<0.05).

## Tissue expression profiles of miRNAs

To gain insight on their biological functions, we investigated the tissue distribution of the 8 miRNAs that were associated with age in a wide array of body tissues collected from calves and included the liver-specific miR-122 [42] as a control (Fig 5). Results showed that none of the miRNAs analyzed, except miR-122, were tissue-specific. However, some were distinctly enriched in certain tissues, for example, miR-142-5p and miR-363 in spleen and lymph nodes, and miR-30c-5p in kidney compared to the other tissues analyzed.

## Prediction of miRNA target gene pathways

To complement tissue expression analyses, we searched for biological pathway targets of the human homologues of the same 8 miRNAs of interest using miRPath. miR-154b could not be included because it did not have a human homologue. Many of the pathways targeted by these miRNAs were found to be involved in metabolism and cell signaling (Table 3).

## Discussion

The health and performance of young cattle may have a large influence on lifetime productivity and hence the profitability of the dairy industry; early first service and first lactation are important to maximize productivity, and poor health and infertility affect the animal's ability to remain in the herd. Yet, robust early-life predictors of health and productivity in cattle are lacking. To address this, we aimed to investigate whether miRNAs are associated with (and could be use as biomarkers of) early life performance traits in calves, including growth, fertility, and disease risk, as well as age, using samples from a total of 91 animals.

Levels of 8 of the miRNAs analyzed in this study changed significantly across early life stages (from calf to lactating cow). Three of these (miR-127, miR-142-5p, miR-30c-5p) corresponded to miRNAs already identified as associated with age in our previous study which did

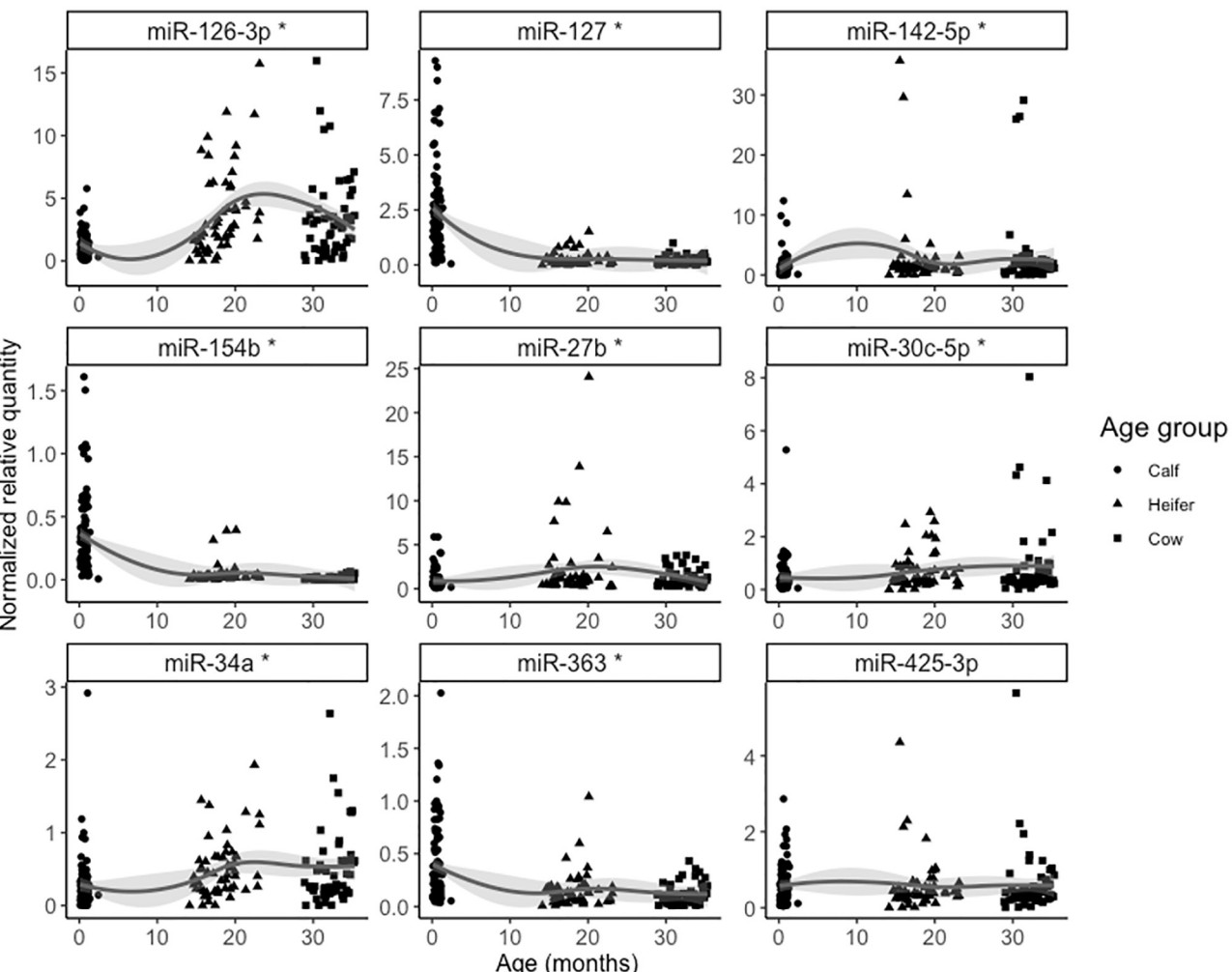

**Fig 3. Loess regression curve of miRNA levels against age (months).** Each miRNA was plotted separately and adjusted for technical variation (batch number). Except for miR-425-3p (FDR-adjusted P = 0.426), all other miRNAs were significantly associated with age (* GLMM: P<0.001; FDR-adjusted).

not involve longitudinal measurements, i.e., within animals [33]. For a further 3 miRNAs (miR-27b, miR-126-3p, miR-154b), the opposite strand or a member of the same miRNA family were associated with age in that study. Also consistent with our previous study [33] was the observation that the most dramatic differences in individual miRNA levels were those between calves and heifers. This is not surprising due to the rapid growth and maturation of different body systems that occurs between birth and puberty.

Among miRNAs whose levels decreased with age, miR-127 displayed the most dramatic changes. A decrease in levels of this miRNA across early life stages has been demonstrated previously in both cattle [33] and pigs [36]. miR-127 is highly expressed in mesenchymal stem cells (MSCs) [44], potentially explaining its broad distribution across the various body tissues analyzed in our study. Previous research has shown that this miRNA inhibits proliferation of MSCs from mouse, rat, and pig, in addition to promoting MSC differentiation towards bone and muscle at the expense of fat [45–48]. These findings point to a role for miR-127 in regulating tissue growth, however, no significant association was found with any of the growth

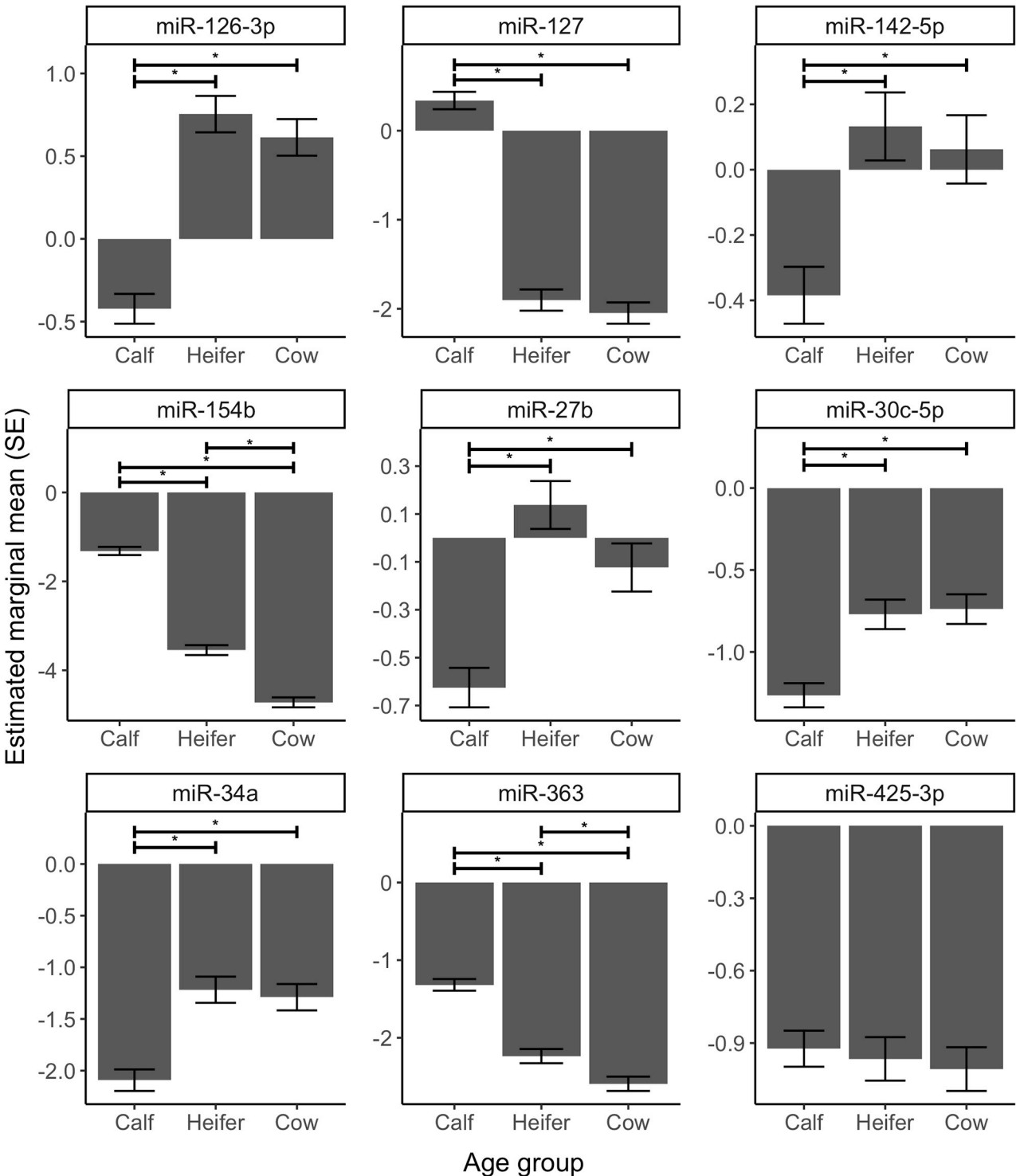

**Fig 4. Estimated marginal means (+/- SE) derived from a model for the effect of age group on miRNA levels.** Significant pairwise comparisons between age groups are indicated above bars as stars (* P<0.05; FDR-adjusted).

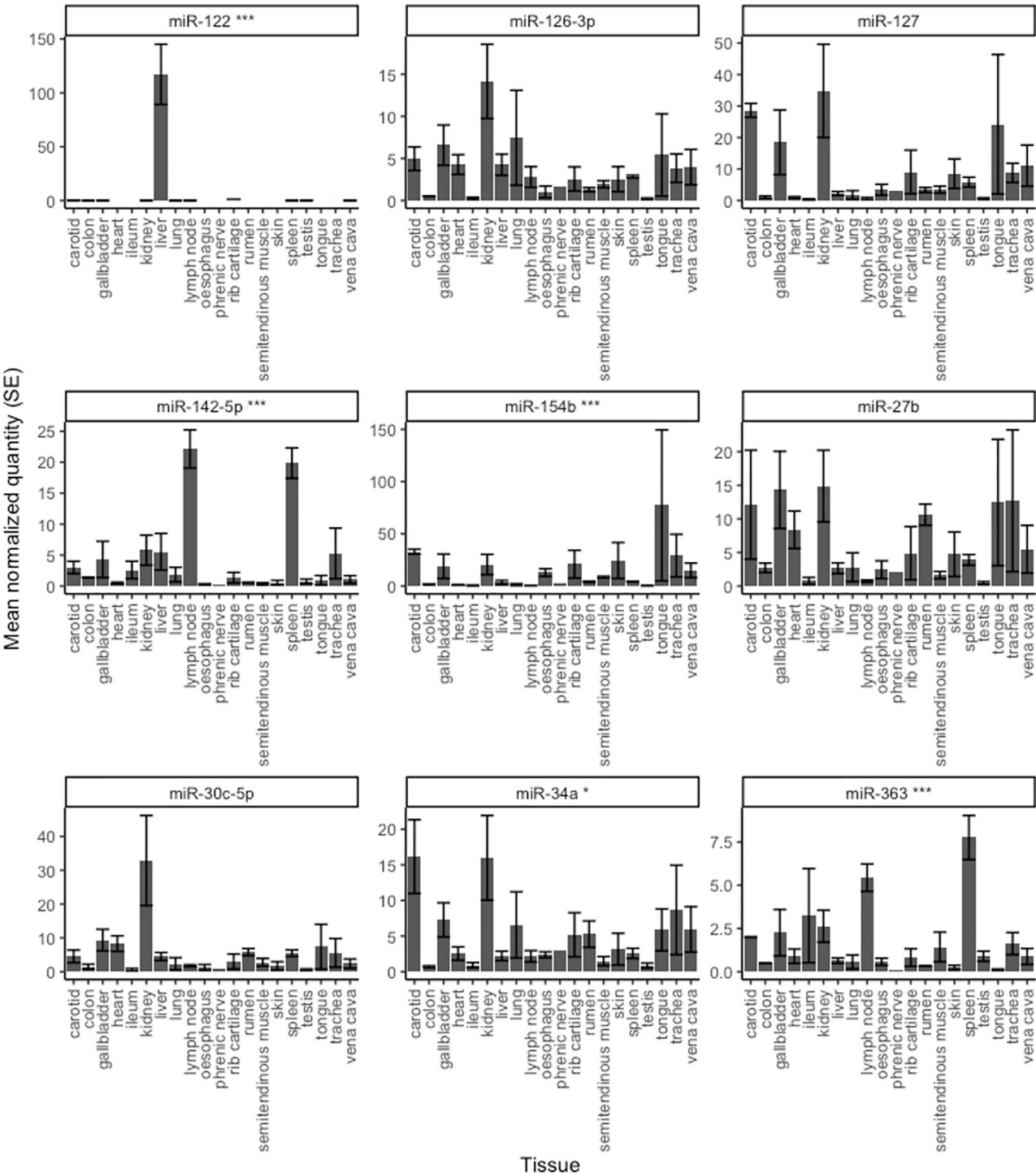

**Fig 5. Mean (+/- SE) levels of age-related miRNAs across body tissues in calves.** Stars indicate the overall effect of tissue type on miRNA expression (Wald tests of GLMM: * P<0.05; *** P<0.001; FDR-adjusted).

**Table 3. KEGG Pathways targeted by human homologues of bovine miRNAs associated with age[1].**

| KEGG pathway | P-value | No. genes | No. miRNAs | miRNAs |
|---|---|---|---|---|
| Fatty acid metabolism | 0.000 | 20 | 4 | miR-27b, miR-30c-5p, miR-34a, miR-363 |
| p53 signaling pathway | 0.000 | 46 | 4 | miR-142-5p, miR-27b, miR-30c-5p, miR-34a |
| Cell cycle | 0.000 | 78 | 4 | miR-142-5p, miR-27b, miR-30c-5p, miR-34a |
| Fatty acid biosynthesis | 0.000 | 4 | 3 | miR-27b, miR-30c-5p, miR-34a |
| Progesterone-mediated oocyte maturation | 0.011 | 47 | 3 | miR-27b, miR-30c-5p, miR-34a |
| Oocyte meiosis | 0.000 | 40 | 2 | miR-27b, miR-30c-5p |
| Mucin type O-Glycan biosynthesis | 0.000 | 11 | 2 | miR-27b, miR-30c-5p |
| Thyroid hormone signaling pathway | 0.000 | 58 | 2 | miR-27b, miR-34a |
| Ubiquitin mediated proteolysis | 0.000 | 44 | 2 | miR-142-5p, miR-30c-5p |
| ECM-receptor interaction | 0.001 | 17 | 2 | miR-127, miR-27b |
| Protein processing in endoplasmic reticulum | 0.004 | 57 | 2 | miR-27b, miR-30c-5p |
| mTOR signaling pathway | 0.021 | 23 | 2 | miR-27b, miR-126-3p |
| TGF-beta signaling pathway | 0.024 | 30 | 2 | miR-27b, miR-30c-5p |
| Biosynthesis of unsaturated fatty acids | 0.047 | 5 | 2 | miR-30c-5p, miR-363 |
| Adherens junction | 0.004 | 31 | 1 | miR-34a |
| Lysine degradation | 0.012 | 10 | 1 | miR-30c-5p |
| Hippo signaling pathway | 0.025 | 30 | 1 | miR-27b |

[1]Disease related pathways have been excluded.

parameters examined in our study. Whether comparing different breeds that markedly differ in growth characteristics and body carcass composition, e.g., dairy vs beef breeds, would provide any such associations still needs to be determined.

Another miRNA that showed a marked decrease in plasma levels with age was miR-154b, a ruminant-specific miRNA. There is no functional data available for miR-154b in the literature. However, miR-154c has previously been shown to be highly expressed in fat compared to other body tissues [49] and to be downregulated in udder tissue during bovine mastitis [50] and in sheep serum after lipopolysaccharide challenge [51], indicating that it may be involved in tissue responses to infection.

miR-363, a miRNA that also decreased with age, has previously been linked to inflammation [52, 53]. In cattle, miR-363 was upregulated during Mycobacterium avium subsp. paratuberculosis infection [54]. We found that this miRNA was elevated in spleen and lymph node, consistent with a role in inflammation. Though we did not identify a significant association between this miRNA and number of infections in early life, we found that it was involved in several ratios that were significantly associated with number of infections or ADG-wean prior to FDR adjustment. A role of this miRNA in the dynamics of inflammation in cattle should be investigated in the future.

Among miRNAs whose plasma levels increased with age, miR-34a showed the largest change during the calf-to-heifer transition. This and other members of the miR-34 family have been linked to inflammatory processes in several species [55–57]. In addition, miR-34a was reported to be upregulated in skeletal muscle of cattle with high residual feed intake [58]. Future target validation studies in cattle should be carried out to clarify whether the changes in miR-34a observed in our study are associated with age-related changes in metabolism or propensity for inflammation.

Plasma miR-126-3p levels also increased markedly in heifers relatively to calves. This is consistent with previous findings in cattle [33] and humans [59]. An increase in levels of this

miRNA in circulation with age has been implicated in endothelial senescence and vascular disease [60]. Moreover, miR-126-3p did not increase with age in diabetes patients, as it did in healthy people [61]. Since miR-126 is intrinsically involved in the regulation of tissue angiogenesis [62] it can be hypothesized that changes in plasma miR-126-3p during growth in cattle may reflect their role in regulating blood vessel development during rapid body growth. This miRNA has also been implicated in lipid synthesis in mouse mammary epithelia, where its levels decreased during gestation and lactation [63]. Further study is required to elucidate the potential role of miR-126-3p in bovine mammary glands.

miR-142-5p is highly expressed in white blood cells [64] and has been linked to inflammation [65]. It has also been linked to lipopolysaccharide challenge responses and mastitis in cattle [32, 66, 67]. Consistent with this, miR-142-5p was elevated in lymph node and spleen relative to other tissues in our study. Thus, the age-related increase in miR-142-5p observed by us may be related to so-called "inflammaging" reported in cattle by Zhang and colleagues [68].

The temporal expression patterns observed for the above miRNAs raise the question of whether some of the observed changes may be related to puberty. miRNAs have been shown to be involved in onset of puberty in mice by regulating neuronal activity at the hypothalamic level [69, 70]. Moreover, changes in circulating miRNA levels have been reported in association with puberty in humans [71]. However, none of the miRNAs associated with age in our study have been linked previously to puberty onset. Furthermore, the wide body tissue distribution of these miRNAs together with functional evidence from previous studies, outlined in the paragraphs above, rather suggest a broader involvement in body growth and metabolism in calves.

Finally, in terms of miRNA associations with animal traits, we found 2 individual miRNAs and 8 ratios between miRNAs that were significantly associated with early performance. The former included associations between miR-126-3p and number of infections, and between miR-30c-5p and ADG until weaning. However, none of those retained significance when p-values were adjusted for multiple testing. Given that only a limited set of the total number of candidate miRNAs identified by PCR arrays were chosen for validation in the current study (9 out 85 miRNAs), and that only early life traits were analyzed, future work should consider additional miRNAs from our list and include later life traits, for example, milking traits, fertility after the first calving, or risk for specific diseases such as mastitis and lameness, which were beyond the scope of this study. For example, miR-127 was not associated with any trait in the present study but has been previously associated with lameness [33].

## Conclusions

Strong associations with age, from calf to first lactation cow (~2 years of age), were identified for 8 miRNAs, namely miR-126-3p, miR-127, miR-142-5p, miR-154b, miR-27b, miR-30c-5p, miR-34a, and miR-363. Existing functional evidence indicates that these miRNAs are involved in regulating tissue growth and metabolism, suggesting they may be involved in early development in cattle. Investigation of additional candidate miRNAs and/or traits to those studied here would be warranted to further elucidate their roles in developmental processes.

## Supporting information

**S1 Data. Ct values derived from PCR arrays of plasma from 12 calves.** NA = not detectable.
(XLSX)

**S1 File.**
(PDF)

## Acknowledgments

We are very thankful to staff at Langhill Farm, particularly Mr. Wilson Lee, as well as to Dr. Tim Connelly, Dr. Thomas Tzelos, Dr. Enguang Rong and Dr. Seungmee Lee for assistance with sample collection. For the purpose of open access, the authors have applied a CC-BY public copyright license to any Author Accepted Manuscript version arising from this submission.

## Author Contributions

**Conceptualization:** Georgios Banos, Francesc Xavier Donadeu.

**Formal analysis:** Madison MacLeay.

**Funding acquisition:** Georgios Banos, Francesc Xavier Donadeu.

**Investigation:** Madison MacLeay.

**Methodology:** Madison MacLeay.

**Resources:** Francesc Xavier Donadeu.

**Supervision:** Georgios Banos, Francesc Xavier Donadeu.

**Validation:** Madison MacLeay.

**Writing – original draft:** Madison MacLeay.

**Writing – review & editing:** Georgios Banos, Francesc Xavier Donadeu.

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
