## [Decision Letter · Decision Letter 0]

14 Apr 2023

PONE-D-23-07452Association of plasma miRNAs with early life performance and aging in dairy cattlePLOS ONE

Dear Dr. Donadeu,

Thank you for submitting your manuscript to PLOS ONE. After careful consideration, we feel that it has merit but does not fully meet PLOS ONE’s publication criteria as it currently stands. Therefore, we invite you to submit a revised version of the manuscript that addresses the points raised during the review process.

We look forward to receiving your revised manuscript.

Kind regards,

Gizat Almaw

Academic Editor

PLOS ONE

Journal Requirements:

Reviewers' comments:

Reviewer's Responses to Questions

**Comments to the Author**

1. Is the manuscript technically sound, and do the data support the conclusions?

Reviewer #1: Yes

Reviewer #2: Yes

2. Has the statistical analysis been performed appropriately and rigorously? 

Reviewer #1: Yes

Reviewer #2: Yes

3. Have the authors made all data underlying the findings in their manuscript fully available?

Reviewer #1: Yes

Reviewer #2: Yes

4. Is the manuscript presented in an intelligible fashion and written in standard English?

Reviewer #1: Yes

Reviewer #2: Yes

5. Review Comments to the Author

Reviewer #1: The study aimed to identify circulating miRNAs associated with early life performance traits and aging in dairy cattle. Six miRNAs differed significantly in calves with poor growth/fertility, and 85 miRNAs were associated with at least one animal trait. Eight miRNAs changed significantly with age and could provide useful biomarkers of aging in cattle.

it is interesting and informative, but need to clarify several things in your manuscript. for example, It would be helpful to indicate whether it is in humans or cattle when stating that miRNAs are natural key mediators that respond to pathophysiological stressors and aging in cells.

have you tried to find miRNA-target genes SNP? you may included relationship between miRNA and genotype if available.

Reviewer #2: In their manuscript titled: "Association of plasma miRNAs with early life performance and aging in dairy cattle" Madison MacLeay and colleagues have looked for the association between circulating miRNA profiles, and early life performance traits in dairy cattle using measurements along the life of the same animal. The main aim of the manuscript is to identify early-life predictors of health and productivity for the benefit of the dairy industry.

The research question is clear and the study addresses an important issue for profitability derived from dairy breed cattle productions.

The methodology is robust as well as the statistical approach. The study is clearly stated.

L 135-137: The reviewer is impressed by the high percentage of dead or culled animals within the 35 months of age. Just over half of the animals were collected twice along the life.

The reviewer didn’t find so coherent the investigation on miRNAs’ expression in different tissues of young male calves. Interesting, but not so harmonic. It could have been better on female of the same breeds during the first month of age.

Results showed that the bigger differences in miRNAs levels were between calves and heifers. Puberty can affect the miRNAs profiles. The reviewer believes that the manuscript could be improved with a discussion about this point.

L 91: changes “were” associated

6. PLOS authors have the option to publish the peer review history of their article (what does this mean?). If published, this will include your full peer review and any attached files.

Reviewer #1: **Yes: **Inchul Choi

Reviewer #2: No

---

## [Author Response · Author response to Decision Letter 0]

22 May 2023

Reviewer #1: 

it is interesting and informative, but need to clarify several things in your manuscript. for example, It would be helpful to indicate whether it is in humans or cattle when stating that miRNAs are natural key mediators that respond to pathophysiological stressors and aging in cells.

Response: Thanks for your positive feedback. We have now indicated the species our statement in Line 87 refers to. Because in general miRNAs are functionally conserved, the statement would be applicable across species.

have you tried to find miRNA-target genes SNP? 

Response: We did not, as our aim was not to identify targets for miRNAs, however, we agree that this would be worth doing in a follow up study. 

you may included relationship between miRNA and genotype if available.

Response: Unfortunately, we do not have genotypes for the animals used in the analyses; we would have done such analyses if those had been available.

Reviewer #2: 

The research question is clear and the study addresses an important issue for profitability derived from dairy breed cattle productions.

The methodology is robust as well as the statistical approach. The study is clearly stated.

Response: Thanks for your comments.

L 135-137: The reviewer is impressed by the high percentage of dead or culled animals within the 35 months of age. Just over half of the animals were collected twice along the life.

Response: Thanks for spotting this. In the original submission, we mistakenly included the number of cows sampled from one of the cohorts only, i.e. 98 cows from the 17/18 season. We have now updated this to include the total number of cows samples from both cohorts (171 cows; Line 137), in line with the information provided for the other two age groups (calves and heifers). We apologise for the oversight. 

The reviewer didn’t find so coherent the investigation on miRNAs’ expression in different tissues of young male calves. Interesting, but not so harmonic. It could have been better on female of the same breeds during the first month of age.

Response: We agree that <1 month-old females would have been ideal for tissue analyses; however, such tissues were not available to us, and we reasoned that samples from slightly older male calves (Holstein Friesian or crosses, same as the calves used for blood analyses) would effectively address our primary question of whether any miRNA would be tissue specific. This is because:

1) Tissue specificity of miRNAs under non-pathological conditions is not expected to change with age, particularly in a short time frame such as considered in our study (1 vs 5 months of age) 

2) Sex-biasing in miRNA expression is limited and it has not shown to involve the miRNAs analysed in our study, e.g. BMC Medical Genomics (2015) 8:61 and J Pediatr 2021;235:138-43.

Thus, we reasoned that expression profiles in tissues from 5-month old male calves provide a good proxy for miRNA tissue specificity in younger animals. We hope that the reviewer will agree that this was a reasonable assumption. Indeed, our tissue expression profiles agree with those in previous studies (BMC Genomics. 2018; 19: 243, and Sci. Data. 2019; 6:190013, providing further confidence in our approach. 

Results showed that the bigger differences in miRNAs levels were between calves and heifers. Puberty can affect the miRNAs profiles. The reviewer believes that the manuscript could be improved with a discussion about this point.

Response: We agree with the reviewer that this would improve our discussion and, accordingly, we have added a new paragraph and 3 additional references to the Discussion (Lines 463-471). 

L 91: changes “were” associated

Response: ‘Was’ has been replaced by ‘were’.

---

## [Editor Report · Decision Letter 1]

26 Jun 2023

Association of plasma miRNAs with early life performance and aging in dairy cattle

PONE-D-23-07452R1

Dear Dr. Donadeu,

We’re pleased to inform you that your manuscript has been judged scientifically suitable for publication and will be formally accepted for publication once it meets all outstanding technical requirements.

Kind regards,

Gizat Almaw

Academic Editor

PLOS ONE
---

## [Editor Report · Acceptance letter]

28 Jun 2023

PONE-D-23-07452R1 

Association of plasma miRNAs with early life performance and aging in dairy cattle 

Dear Dr. Donadeu:

I'm pleased to inform you that your manuscript has been deemed suitable for publication in PLOS ONE. Congratulations! Your manuscript is now with our production department. 

Kind regards, 

on behalf of

Dr. Gizat Almaw 

Academic Editor

PLOS ONE